# Insights into Protective Effects of Different Synbiotic Microcapsules on the Survival of *Lactiplantibacillus plantarum* by Electrospraying

**DOI:** 10.3390/foods11233872

**Published:** 2022-11-30

**Authors:** Shu-Fang Li, Kun Feng, Ru-Meng Huang, Yun-Shan Wei, Hong Wu

**Affiliations:** 1Guangdong Province Key Laboratory for Green Processing of Natural Products and Product Safety, School of Food Science and Engineering, South China University of Technology, Guangzhou 510640, China; 2Henan Key Laboratory of Cold Chain Food Quality and Safety Control, College of Food and Bioengineering, Zhengzhou University of Light Industry, Zhengzhou 450001, China

**Keywords:** probiotics, FOS, fish oil, coaxial electrospraying, encapsulation, viability

## Abstract

This study evaluated the protective effects of different synbiotic microcapsules on the viability of encapsulated *Lactiplantibacillus plantarum* GIM1.648 fabricated by electrospraying. The optimum amount of substrate for three synbiotic microcapsules separately containing fructooligosaccharide (FOS), fish oil, and the complex of both were 4% FOS (SPI-F-L-P), 20 μL fish oil (SPI-O-L-P) and the complex of 20 μL fish oil, and 2% FOS (SPI-O-F-L-P), respectively. The obtained synbiotic microcapsules had a better encapsulation efficiency (EE) and survival rate (SR) after in vitro digestion than microcapsules without the addition of substrate (SPI-L-P) and SPI-O-F-L-P presented the highest EE (95.9%) and SR (95.5%). When compared to SPI-L-P, the synbiotic microcapsules possessed a more compact structure as proved by the SEM observation and their cell viability were significantly improved in response to environmental stresses (heat treatment, freeze drying, and storage). The synbiotic microcapsules containing the complex of FOS and fish oil showed the best beneficial effect, followed by ones with fish oil and then FOS, suggesting the FOS and fish oil complex has more potential in application.

## 1. Introduction

Probiotics are living microorganisms that, when administered in adequate amounts, confer health benefits on the host [1]. They can prevent or treat certain diseases through the enhancement of the gut barrier function, the production of antimicrobial compounds, and the modulation of host immune responses [2]. Hence, probiotics products, including food and dietary supplements, have attracted the widespread attention of consumers. In order to confer their potential effects, a sufficient number of probiotics must be metabolically active to reach the colon of the gastrointestinal tract [3,4]. However, the viability of probiotics is inevitably damaged by surrounding stresses in the processing, storage, interaction with food matrix, and movement through the digestive tract [5]. Therefore, encapsulation to support the survival of probiotics against negative circumstances has been proposed and proven to be an efficient approach [6].

Nowadays, multilayer structures composed of an inner core and two or more shell layers have gained more attention because they can enhance the resistance of probiotics to adverse conditions and achieve their colon-targeted release. The encapsulation technologies, including emulsion, co-extrusion, layer by layer deposition, and coaxial electrospraying, are commonly used for designing multilayer structures. Among them, coaxial electrospraying is a more promising encapsulation technology with several advantages [7]. Firstly, the experimental setup is simple and low cost, which can produce nearly mono-dispersed microcapsules with high reproducibility under a one-step model [8]. Secondly, the whole process can be operated in an aqueous solution without any organic solvents at room temperature, which is favorable to maintain the viability of probiotics. In addition, the biopolymers for probiotic encapsulation are generally biodegradable, biocompatible, and approved for use in the food industry, such as sodium alginate (SA), pectin (PEC), and soy protein isolate (SPI). 

When active compounds such as prebiotics and polyunsaturated fatty acids (PUFAs) are co-encapsulated with probiotics, the viability of probiotics are further improved due to the synergistic benefits. The mixture of living microorganisms and substrate(s) selectively utilized by host microorganisms that confer health benefits on the host is referred to as synbiotic [9,10] and the functional foods containing synbiotic can significantly contribute to health by regulating gut microbiota and improving immune response [11,12]. Up to now, low molecular weight oligosaccharides have been extensively focused as typical prebiotics for their beneficial role, especially fructooligosaccharide (FOS), galactooligosaccharide (GOS), and inulin. For example, FOS can improve the viability of lactobacilli strains both for the process of digestion and storage at 4 °C [13]; the FOS, GOS, and inulin were, respectively, co-encapsulated with probiotics to enhance their survivability during gastrointestinal digestion and longtime storage [14,15]. In recent years, polyunsaturated fatty acids, such as omega-3 fatty acids (i.e., EPA and DHA), are considered as potential prebiotics [16]. Fish oil contains a high proportion of PUFAs with numerous physiological roles [17]. We also found that the incorporation of fish oil into microencapsulation system could enhance the survivability of *Lactiplantibacillus plantarum* GIM1.648 during digestion, processing, and storage [18]. However, as far as we know, there are few works to compare the beneficial effects of oligosaccharide and fish oil and the protective effect of the combination of oligosaccharide and fish oil has not been reported yet.

Therefore, in order to gain insights into the effects of different types of substrates on improving the survivability of probiotics encapsulated in synbiotic microcapsules under harsh conditions, in this study, *Lactiplantibacillus plantarum* GIM1.648 (*L. plantarum* GIM1.648) was served as a model probiotic and a series of synbiotic microcapsules including *L. plantarum* GIM1.648 and different types of substrates (FOS, fish oil, the complex of FOS, and fish oil) were prepared by coaxial electrospraying. Firstly, the optimum additive amount of substrate was determined in terms of the encapsulation efficiency and survival rate of encapsulated probiotics after transiting through the simulated gastrointestinal fluids. Subsequently, the characterization of microcapsules was studied. Finally, the survivability of probiotics encapsulated in different synbiotic microcapsules was investigated following heat treatment, freeze drying, and storage.

## 2. Materials and Methods

### 2.1. Materials

*L. plantarum* GIM1.648 was a gift from Prof. Jiguo Yang’s laboratory belonging to the School of Food Science and Engineering, South China University of Technology, and it was purchased from CINOBIO-TEC (Shanghai, China). MRS (de Man Rogosa Sharpe) broth and agar were supplied by Huankai Microorganism Technology Co., Ltd. (Guangzhou, China). The FOS and fish oil were, respectively, purchased from Xinjinshan Biotechnology Co., Ltd. (Yunfu, China) and Sinomega Biotechnology Co., Ltd. (Zhoushan, China). The SPI, pectin, and pectinase were bought from Yuanye Biological Technology Co., Ltd. (Shanghai, China). The sodium alginate (SA, from brown algae) was bought from Sigma-Aldrich company (Shanghai, China). The pepsin (3000 U/mg), trypsin (250 U/mg), and bile salt were supplied from Aladdin Biological Technology Co., Ltd. (Shanghai, China).

### 2.2. Cell Culture

A thawed solution of *L. plantarum* GIM1.648 was inoculated into 40 mL fresh MRS broth and activated twice at 37 °C for 18 h, as previously described [19]. After that, the culture located at the stationary phase was centrifuged at 4 °C and 5000 rpm for 5 min to obtain the cell pellet and then it was washed twice and suspended in sterile normal saline solution (0.85%, *w*/*v*) for later use. The counting of *L. plantarum* GIM1.648 was performed using the plate colony-counting method, and the cell counts were expressed as a log of colony forming units (CFU) per milliliter solution.

### 2.3. Preparation of Polymer Solutions

The solutions of SA (2.5%, *w*/*v*), PEC (2%, *w*/*v*), and FOS (1%, 2%, 3%, 4%, 5%, *w*/*v*) were obtained by, respectively, weighing and dissolving each powder into the sterile water. The SA (2.5%, *w*/*v*) and PEC (2%, *w*/*v*) solution was mixed in a ratio of 6:8 (*w*/*w*) as a shell layer solution. A blank core solution consisting of SA (2.5%, *w*/*v*) and 100 uL of cell pellet suspension (10^10^ CFU/mL) was served as a control. The core solutions including FOS were prepared by adding different concentrations of FOS (1–5%) into 5 sterilized tubes containing a blank core solution and mixing sufficiently. The mixture of fish oil (20 μL) and a blank core solution was served as a core solution containing fish oil. To prepare the core solutions involving FOS and fish oil complex, FOS (2%, 3%, 4%), fish oil (10 μL, 15 μL, 20 μL), and the blank core solution were added into 9 sterile tubes, respectively. The SPI (3%, *w*/*v*) solution was prepared by dissolving SPI powder into sterile water under magnetic stirring and the receiving fluid was obtained by adding CaCl_2_ (1.5%, *w*/*v*) powder into SPI (3%, *w*/*v*) solution and stirred at an ambient temperature. 

### 2.4. Production of Microcapsules by Coaxial Electrospraying

The coaxial electrospraying setup consisted of a voltage power supply (ES50P-5W/DAM, Gamma, Ormond Beach, FL, USA), two syringe pumps (NE-300, New Era Pump Systems Inc., Farmingdale, NY, USA), a homemade concentric needle with inner diameters of 1.07 mm (outer needle of 17 gauge) and 0.5 mm (inner needle of 21 gauge), and a collector. The core solution and shell solution were individually pumped from a sterile 5 mL plastic syringe pass through the inner and outer needle at the rate of 3 mL/h and 7 mL/h. The needle tip was located at 10 cm from the surface of receiving fluid and a voltage of 12 kV was applied [18]. The microcapsules were formed when the droplets through from concentric needle dropped vertically into the receiving fluid. The room temperature was about 25 °C and the relative humidity was in the range of 40–50%, respectively.

The obtained synbiotic microcapsules were named in view of the fact that they contained probiotics and distinct types of substrates. The microcapsules separately containing FOS, fish oil, and the complex of FOS and fish oil were referred to as SPI-F-L-P, SPI-O-L-P, and SPI-O-F-L-P, respectively, and the blank microcapsule without additional substrate was called SPI-L-P.

### 2.5. Encapsulation Efficiency

The method for measuring the encapsulation efficiency (EE) of *L. plantarum* GIM1.648 in microcapsules was dependent on our published study [18]. Finally, the EE of *L. plantarum* GIM1.648 was calculated on the basis of Equation (1):EE (%) = (N/N_0_) × 100(1)
where N is the number of viable cells released from microcapsules (log CFU/mL) and N_0_ is the initial number of viable cells (log CFU/mL) added to the core solution. 

### 2.6. Survival Rate of Encapsulated L. plantarum GIM1.648 Exposed to Simulated Digestive System

The survival rate (SR) of the encapsulated cells was calculated after passing sequentially through the gastric fluid, intestinal fluid, and colonic fluid. The simulated gastrointestinal fluid was prepared on the basis of the previous study with slight modifications [20]. The pepsin (3.3 mg/mL) was added into sterilized normal saline (0.85%, *w*/*v*, pH 2.0) to obtain a simulated gastric fluid (SGF). The simulated intestinal fluid (SIF) was obtained by dissolving bile salt (3 mg/mL) and trypsin (1 mg/mL) into a sterilized saline solution (0.85%, *w*/*v*, pH 6.8). The simulated colonic fluid (SCF) was also prepared by adding pectinase (0.1 mg/mL) to a sterilized phosphate-buffered solution (0.1 M, pH 7.4). All the fluids were sterilized by filtering with 0.22 μm sterile membrane filters and pre-warmed at 37 °C before use.

The process of microcapsules exposed to the digestive system was in accordance with our previous work [18]. The SR of probiotics was calculated based on Equation (2):SR (%) = (N/N_0_) × 100(2)
where N is number of cells (log CFU/mL) released from microcapsules after sequential incubation in simulated gastrointestinal fluids; N_0_ is the number of cells (log CFU/mL) before exposure to simulated gastrointestinal fluids. 

### 2.7. Characterization of Microcapsules

The surface morphology of microcapsules and the internal structure of these microcapsules frozen and sectioned with liquid nitrogen were determined by scanning electron microscopy (SEM, EVO18, Oberkochen, Germany) at an accelerating voltage of 10 kV. The samples were fixed on pieces of aluminum stubs with double-side adhesive carbon tape and observed after Pt sputtering using a sputter coater (K550, Emitech, Ashford, UK) under vacuum.

Fourier transform infrared spectroscopy (FTIR) spectra of different samples (SPI, PEC, SA, FOS, fish oil, *L. plantarum* GIM1.648, and various microcapsules) were conducted by a FTIR spectroscopy (Bruker Model Equinox 55, Ettlingen, Germany) in the spectra range of 4000–500 cm^−1^ with 4 cm^−1^ resolution to investigate their chemical interactions and structure characteristics.

The crystal structure of different samples (SPI, PEC, SA, FOS, fish oil, *L. plantarum* GIM1.648, and various microcapsules) were evaluated by using an X-ray diffraction (XRD) instrument (Malvern Panalytical B.V., Almelo, The Netherlands). The samples were analyzed at 12°/min from 5–60° of 2θ angle by employing Cu-Kα radiation.

The thermogravimetric analysis (TGA) of various microcapsules were studied by means of a thermal analyzer (STA449F3, Selb, Germany) according to the degradation behavior of samples, which were operated at the temperature range from 30 °C to 600 °C at a fixed heating rate of 10 °C/min under nitrogen atmosphere. 

### 2.8. Tolerance of Encapsulated L. plantarum GIM1.648 to Adverse Conditions

#### 2.8.1. Heat Treatment

The 0.5 g of various wet microcapsules and 0.5 mL free *L. plantarum* GIM1.648 were separately added to sterile tubes and then they were incubated in water baths at different temperatures of 50 °C, 60 °C, and 70 °C for 30 min [21]. After incubation, the probiotics were released from microcapsules and enumerated on MRS agar as described in Section 2.5.

#### 2.8.2. Freeze Drying

The 0.5 g of various wet microcapsules and 0.5 mL free *L. plantarum* GIM1.648 were separately dried using a freeze-dryer (Gamma 1–16/2–16 LSC, Marin Christ, Osterode, Germany). The viable cells in the microcapsules both before and after freeze-drying were measured on the basis of the method of Section 2.5.

#### 2.8.3. Storage Study

The 0.5 g of various wet microcapsules and 0.5 mL free *L. plantarum* GIM1.648 were separately transferred into sterile tubes and stored at 4 °C and 25 °C for 15, 30, 60, and 90 days in dark airtight vials [22]. The number of viable cells released from samples were measured as described in Section 2.5.

### 2.9. Statistical Analysis

All the experiments were performed in triplicate and the data were expressed as mean ± standard deviation. The data were subjected to the analysis of variance (ANOVA) using SPSS statistics 24.0, followed by the LSD and Duncan tests. The statistical significance level was considered when *p* < 0.05.

## 3. Results and Discussion

### 3.1. Effects of Synbiotic Microcapsules Containing Different Substrates on the Encapsulation Efficiency and Survival Rate of L. plantarum GIM1.648

It has been demonstrated that the amount of substrate in synbiotic microcapsules is an important factor to influence the viability of probiotics [21]. We had found that the amount of 20 μL fish oil was optimal to prepare SPI-O-L-P microcapsules, which showed the highest EE (95.1%) and SR (94.8%) after passing through the simulated fluids [18]. To objectively evaluate the effect of distinct substrates on the viability of encapsulated probiotics in synbiotic microcapsules, it was necessary to optimize the levels of FOS and the complex of FOS and fish oil. 

Figure 1a depicted the effect of the FOS concentration on the EE of encapsulated probiotics. It was observed that the EE of *L. plantarum* GIM1.648 significantly increased with the increase in FOS concentration and reached the highest value of 92.8% at 4% FOS (*p* < 0.05). The further increase in FOS concentration above 4% did not significantly change the EE (*p* > 0.05). This might be that the addition of FOS caused the structure of microcapsules to be more rigid to protect probiotics. A similar study had encapsulated *Lacticaseibacillus rhamnosus* with SA and different concentrations of FOS by extrusion technique and the EE of cells was found to vary from 84% to 91% [23]. The introduction of FOS had an obvious effect on the survivability of probiotics in the process of in vitro digestion (Figure 1b). The SR of encapsulated *L. plantarum* GIM1.648 increased when the FOS concentration ranged from 0 to 4% and reached the maximum of 93.8% at 4%. This possible reason was that FOS could fill the pores of microcapsules to prevent the diffusion of simulated gastrointestinal fluids into them; on the other hand, FOS could act as an energy source that was metabolized by probiotics to maintain their growth throughout the process. This result was consistent with the previous report [14], in which they produced *L. acidophilus* microcapsules containing Hi-maize by external ionic gelation and found that the addition of Hi-maize resulted in a higher count of viable cells after being exposed to simulated gastrointestinal conditions. Hence, 4% FOS was considered to be optimal for preparation of the SPI-F-L-P microcapsules.

Nine samples of microcapsules containing different levels of FOS and fish oil were prepared. Figure 1c,d described the protective effects of different formulations on the EE and SR of *L. plantarum* GIM1.648. Obviously, the EE and SR of probiotics encapsulated in the microcapsules containing FOS and the fish oil complex were higher than that without additional substrates (*p* < 0.05) and the highest values of 95.9% and 95.5% for the EE and SR were obtained at the eighth sample, which included 2% FOS and 20 µL fish oil. It had also been reported that the alginate capsules containing the complex of inulin (1%, *w*/*w*) and ascorbic acid (0.5%, *w*/*w*) greatly enhanced the viability of Bifidobacteria in simulated gastrointestinal fluids [24]. Therefore, a combination of 2% FOS and 20 µL fish oil was identified as the best formulation to prepare the SPI-O-F-L-P microcapsules.

The EE and SR of *L. plantarum* GIM1.648 encapsulated in the synbiotic microcapsules containing the optimal amount of distinct substrate were compared and the results were shown in Figure 2. The EE of probiotics encapsulated in the SPI-L-P, SPI-F-L-P, SPI-O-L-P, and SPI-O-F-L-P were 89.1%, 92.8%, 95.1%, and 95.9%, respectively. It indicated that addition substrates significantly enhanced the EE of cells (*p* < 0.05), but there was no significant difference between SPI-O-L-P and SPI-O-F-L-P (*p* > 0.05). In terms of the SR of cells after in vitro digestion, the corresponding values for the synbiotic microcapsules of SPI-F-L-P, SPI-O-L-P, and SPI-O-F-L-P were 92.5%, 93.8%, and 95.5%, respectively. Significant differences were found between the control group of SPI-L-P and three types of synbiotic microcapsules (*p* < 0.05), but no significant difference was observed between SPI-F-L-P and SPI-O-L-P (*p* > 0.05). These results indicated that the beneficial effect of fish oil was greater than or equal to FOS and the FOS and fish oil complex had the most outstanding performance. This phenomenon might be due to the synergistic effect between FOS and fish oil [25].

### 3.2. Characterization of Microcapsules

#### 3.2.1. SEM

The SEM images of the surface and internal morphology of the various microcapsules were shown in Figure 3. It was found that the structure of SPI-L-P was irregular, winkled, and with a broken and pitted surface (Figure 3a). However, the three synbiotic microcapsules exhibited a less spherical shape with distinct surface characteristics. Compared to SPI-L-P, SPL-F-L-P, and SPL-O-L-P, they provided a smoother and more compact surface with less wrinkles and dents. The previous study also reported that the inclusion of FOS to the alginate–gelatin matrix formed a more interconnected network and reduced its pore size [26]. Similarly, Vaziri et al. found that the microcapsules containing *L. plantarum* and DHA-rich oil showed smoother surfaces with less concavities [27]. The possible reason was that the addition of FOS improved the polymeric distribution and helped to fill the pores of microcapsules; in addition, the incorporation of fish oil could reduce the water evaporation and increase the moisture content in microcapsules [17,23]. For SPI-O-F-L-P, it presented the most smooth and compact surface with the least dents among the samples (Figure 3g). The synergistic effect between the fish oil and FOS might account for this. From the SEM images of the cross section of all microcapsules (Figure 3), bacterial cells were obviously observed, but there were no cells on the surface of microcapsules, which meant that they were fully covered inside of the microcapsules.

#### 3.2.2. FTIR

The spectra of wall materials, active compounds, free cells, and different microcapsules were presented in Figure 4. In the spectra of SA powder, the peaks at 3463.19 cm^−1^, 1622.13 cm^−1^, 1416.71 cm^−1^, and 1029.98 cm^−1^ were attributed to O-H stretching, asymmetric and symmetric stretching vibration of COO^−^, and stretching variations of C-C and C-O monosaccharide skeleton [19]. The spectra of the PEC powder showed the peaks at 3428.4 cm^−1^, 1642.38 cm^−1^, 1406.1 cm^−1^, and 1026.13 cm^−1^ belonging to O-H, COO^−^, C-O, and C-O-C groups, respectively. FOS exhibited distinct bands at 2930.83 cm^−1^, 1649.13 cm^−1^, 1457.02 cm^−1^, and 1054.09 cm^−1^, which were stretching and deforming the oligosaccharides [21]. The characteristic absorption peaks at 1655.89 cm^−1^ (amide Ⅰ) and 1538.23 cm^−1^ (amide Ⅱ) of SPI were attributed to the tensile vibration of the C=O stretching and N-H bending. The characteristic peaks of SA, PEC, and SPI were observed in all microcapsules. The regions of amide Ⅰ and amide Ⅱ in all microcapsules exhibited a slight shift compared to SPI, suggesting that the structure of the protein had undergone modification to form a more ordered structure due to the interaction of the protein and polysaccharide, in which the carboxyl groups of polymers interacted with the amino groups of SPI to form the amide group [28]. In addition, the pick intensities of both hydroxyl and carboxyl groups were clearly higher as seen from the spectra of all microcapsules, showing that the cells were encapsulated into the matrix. Overall, the spectra of all microcapsules looked similar in shape, but there were slight differences in the several band intensities. The spectra of fish oil showed several peaks at 3013.77 cm^−1^ (C–H stretching of *cis*-alkaline –HC=CH–), 2929.87 cm^−1^ (C-H stretching of CH_3_ groups), 1737.86 cm^−1^ (C=O of functional groups), and 1159.22 cm^−1^ (C-H stretching of CH_2_ groups) [27]. In the microcapsules of SPI-O-L-P and SPI-O-F-L-P, the unique peaks of fish oil were vanished, which was probably due to its very low amount. In addition, in the spectra of SPI-F-L-P and SPI-O-F-L-P microcapsules, the peaks for O-H stretching were sharper perhaps owing to the presence of FOS. Various studies showed that the absorption peaks of bacterial proteins and nucleic acids are mainly located in the range of 950 to 1200 cm^−1^, which is in line with our results [29,30]. The absorption peaks of cells were found in all microcapsules, further demonstrating that the probiotics were encapsulated in microcapsules. In addition, the absorption peaks of SPI-O-F-L-P were the sharpest among the microcapsules, indicating that SPI-O-F-L-P contained the highest number of probiotics [21].

#### 3.2.3. XRD

The XRD patterns of different samples were depicted in Figure 5. The SPI exhibited board peaks at 9° and 21° and FOS presented broad peaks at 12° and 23°, demonstrating their amorphous structure with a crystalline region [21,31]. A sharp crystalline peak was observed at 13.4° in the pattern of SA, which was reported previously as a typical peak of SA [32]. However, these characteristic peaks almost disappeared in XRD patterns of various microcapsules, which can be attributed to the formation of amorphous complex between polysaccharide and protein by intermolecular interactions, causing the intensity of diffraction peaks to decrease and broadening the patterns. From the XRD pattern of *L. plantarum* GIM1.648, it showed wide diffraction peak at 21°, but the peak intensity progressively reduced in different microcapsules, indicating that bacterial cells were trapped into microcapsules in an amorphous state. The XRD curves of different microcapsules were similar in shape, whereas SPI-O-F-L-P showed a greater peak intensity at 21° than other synbiotic microcapsules, probably due to the SPI-O-F-L-P containing more probiotics.

#### 3.2.4. TGA

Figure 6 shows that all the microcapsules had three thermal degradation stages. There were little differences for all samples during whole thermal degradation stages. The temperatures at a maximum weight loss rate of different microcapsules were 263.8 °C for SPI-L-P, 276.8 °C for SPI-F-L-P, 277.0 °C for SPI-O-L-P, and 282.4 °C for SPI-O-F-L-P. Meanwhile, the percentages of the final residual weight of SPI-L-P, SPI-F-L-P, SPI-O-L-P, and SPI-O-F-L-P were 40.42%, 42.96%, 44.98%, and 47.12%, respectively. These results suggested that the presence of substrates improved the thermal stability of probiotics in synbiotic microcapsules. SPI-O-F-L-P had the best thermal stability among the three synbiotic microcapsules, probably because of the incorporation of FOS and the fish oil complex caused more compact and dense structure of microcapsules.

### 3.3. Tolerance of Encapsulated L. plantarum GIM1.648 to Adverse Conditions

#### 3.3.1. Heat Treatment 

Figure 7a displayed the thermal tolerance of free and encapsulated cells after heat treatment at 50 °C, 60 °C, and 70 °C for 30 min. Apparently, the cell viability of all samples decreased with the increase in treatment temperature and the higher the temperature, the greater the loss of cell viability. However, compared to free cells, the encapsulated cells showed better heat resistance and the thermal stability of different microcapsules followed by a sequence of SPI-O-F-L-P > SPI-F-L-P ~ SPI-O-L-P > SPI-L-P. For example, after treatment at 70 °C, the free cells showed a reduction of 5.92 log CFU/mL, but the corresponding values were 2.56, 1.87, 1.71, and 1.60 log CFU/mL for SPI-L-P, SPL-F-L-P, SPL-O-L-P, and SPI-O-F-L-P, respectively. The heat resistance of SPI-F-L-P and SPI-O-L-P were significantly higher when compared to SPI-L-P (*p* < 0.05), but there was no statistical change between them. Among all the synbiotic microcapsules, the SPI-O-F-L-P maintained the highest count of viable cells (7.37 log CFU/mL), which was significantly higher than the others (*p* < 0.05). Overall, these results suggested that encapsulation is an important way to improve the heat resistance of probiotics. Moreover, the presence of substrates, especially the complex of FOS and fish oil, could significantly enhance the thermal stability of encapsulated probiotics, which were consistent with the above TGA results. It was probably because the synbiotic microcapsules possessed more compact and dense structures, which effectively slow down or isolate the effects of heating on probiotics [33].

#### 3.3.2. Freeze Drying Treatment

During the process of freeze drying, the formation of ice crystals and dehydration stresses cause the structural damage of cells and then the loss of their functions [34]. Figure 7b showed the viability of probiotics in free and encapsulated forms before and after freeze-drying. It can be seen that free cells remained only 3.70 log CFU/mL after freeze drying, which was decreased by 6.07 log CFU/mL when compared to its initial number. While for the bacterial cells encapsulated in various microcapsules, there were only small reductions in viability as indicated by 0.99, 0.81, 0.69, and 0.62 log CFU/mL for SPI-L-P, SPI-O-L-P, SPI-O-F-L-P, and SPI-F-L-P, respectively, indicating the encapsulation could significantly improve the stability of *L. plantarum* GIM1.648 during the process of freeze drying. Compared to SPI-L-P, SPI-F-L-P, and SPI-O-L-P were more efficient and presented a statistical improvement (*p* < 0.05), but no statistical difference between them was detected. SPI-O-F-L-P exhibited the most outstanding performance among the three synbiotic microcapsules, which maintained the highest number of viable cells and showed a significant difference compared to SPI-F-L-P and SPI-O-L-P (*p* < 0.05). The synergic effect between FOS and fish oil may be responsible for this result. On one hand, FOS could act as a cryoprotectant [35]; on the other hand, the structure of microcapsules become more compressed and condensed.

#### 3.3.3. Storage Stability

Storage stability is essential when it comes to the industrial production of probiotics products, that is, the viable cells count should not be lower than 6 log CFU/mL while being consumed [14,36]. Table 1 described the survivability of *L. plantarum* GIM1.648 encapsulated in different microcapsules at 4 °C and 25 °C for the period of storage and free cells were used as the control. When stored at 4 °C for 90 days, the viability of free cells decreased by 4.66 log CFU/mL, whereas the corresponding values for those encapsulated in SPI-L-P, SPI-F-L-P, SPI-O-L-P, and SPI-O-F-L-P were only 1.42, 1.16, 1.01, and 0.89 log CFU/mL, respectively. These results indicated that the presence of substrates, especially the complex of FOS and fish oil, significantly increased the cell survivability during storage at 4 °C (*p* < 0.05). Poletto et al. also reported that the addition of rice bran and inulin into microcapsules enhanced the count of living cells when stored at 7 °C for a longer period [14]. When stored at 25 °C for 90 days, all the samples underwent a significant decline in cell viability and the number of viable cells encapsulated in various microcapsules decreased by approximately 3 log CFU/mL. The previous study also found that the viability of *Lactobacillus acidophilus* declined markedly in microcapsules stored at 25 °C as compared to those stored at 4 °C [37]. It is worth noting that the presence of fish oil negatively affected the viability of encapsulated probiotics when stored at 25 °C for 90 days. The possible reason for this was the poor oxidation stability of fish oil at 25 °C and produced oxidation products in the microcapsules, which reduced the viability of encapsulated probiotics [18,38]. It can be inferred from the above results that the encapsulated probiotics had a better storage stability at 4 °C and the presence of substrates in synbiotic microcapsules further improved the viability of encapsulated cells. Among the three synbiotic microcapsules containing different types of substrates, those with the complex of FOS and fish oil exhibited the most outstanding protective performance for probiotics at a low temperature, which was probably because the resulting microcapsules had more compact and dense structures due to the synergistic effect of FOS and fish oil; on the other hand, the metabolism of probiotics became slow [22] and the fish oil was not easily oxidized at 4 °C [39].

## 4. Conclusions

The multilayer synbiotic microcapsules containing *L. plantarum* GIM1.648 and different types of substrates (FOS, fish oil, and the complex of fish oil and FOS) were successfully fabricated using one-step coaxial electrospraying. The optimum amount of substrate was 4% FOS for SPI-F-L-P, 20 μL fish oil for SPI-O-L-P, and the complex of 20 μL fish oil and 2% FOS for SPI-O-F-L-P. The encapsulation efficiency of probiotics and the survival rate of probiotics after passing through the digestive system were statistically enhanced due to the presence of substrates in synbiotic microcapsules. Among the three synbiotic microcapsules, SPI-O-F-L-P presented the highest encapsulation efficiency (95.9%) and survival rate (95.5%). The SEM observation demonstrated that probiotics were encapsulated in the microcapsules and the structures of synbiotic microcapsules became more compact. The synbiotic microcapsules offered an improved thermal stability confirmed by TGA results. When exposed to environmental stresses including heat treatment, freeze drying, and storage, the survival of probiotics in the synbiotic microcapsules were significantly enhanced in comparison with SPI-L-P. Among the three synbiotic microcapsules containing different types of substrates, the ones containing a complex of FOS and fish oil showed the most outstanding beneficial effect, followed by those with fish oil and then those with FOS, and the synergistic effect between FOS and fish oil may account for this. The results suggest that the addition of substrates into synbiotic microcapsules, whether in individual or combined forms, can significantly improve the resistant ability of encapsulated cells to harsh conditions and that the combination form has more potential to offer better protection.

## Figures and Tables

**Figure 1 foods-11-03872-f001:**
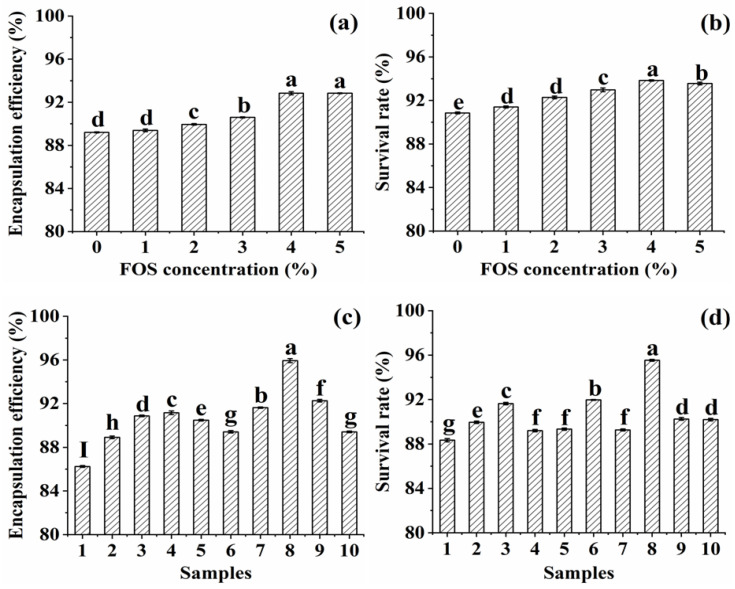
Effects of FOS (**a**,**b**) and the complex of fish oil and FOS (**c**,**d**) on the EE (**a**,**c**) and SR (**b**,**d**) of probiotics. Different lowercase letters above the column indicate a significant difference between samples (*p* < 0.05). (Samples 1: no addition substrate, samples 2–4: 10 µL fish oil + 2% or 3% or 4% FOS; samples 5–7: 15 µL fish oil + 2% or 3% or 4% FOS; samples 8–10: 20 µL fish oil + 2% or 3% or 4% FOS.).

**Figure 2 foods-11-03872-f002:**
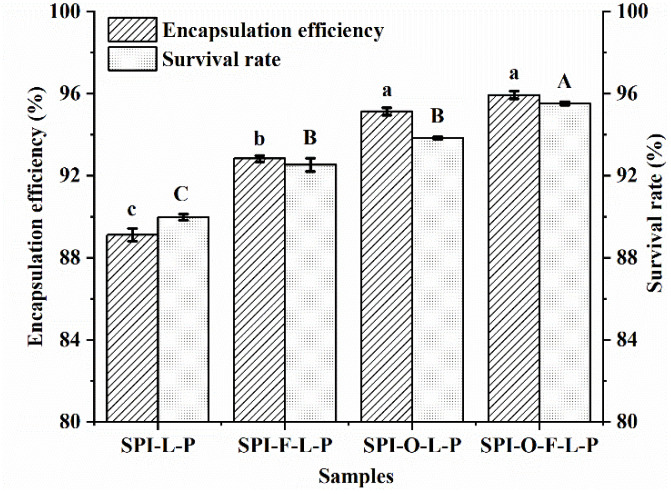
Effects of different substrates at their optimal amounts on the EE and SR of probiotics encapsulated in SPI-L-P, SPI-F-L-P, SPI-O-L-P, and SPI-O-F-L-P. Different lowercase letters denote significant differences of EE between samples and different uppercase letters represent significant differences of SR between samples (*p* < 0.05).

**Figure 3 foods-11-03872-f003:**
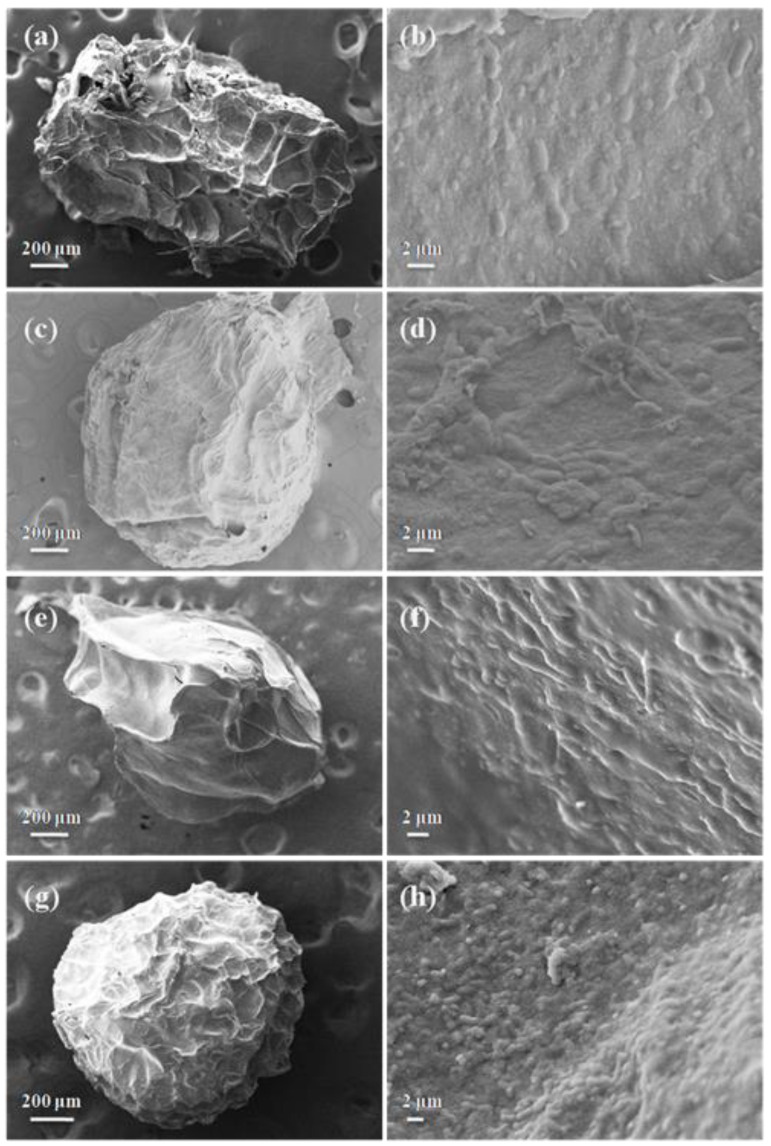
SEM photographs of surface morphology (**a**,**c**,**e**,**g**) and internal structure (**b**,**d**,**f**,**h**) of different microcapsules of SPI-L-P (**a**,**b**), SPI-F-L-P (**c**,**d**), SPI-O-L-P (**e**,**f**), and SPI-O-F-L-P (**g**,**h**).

**Figure 4 foods-11-03872-f004:**
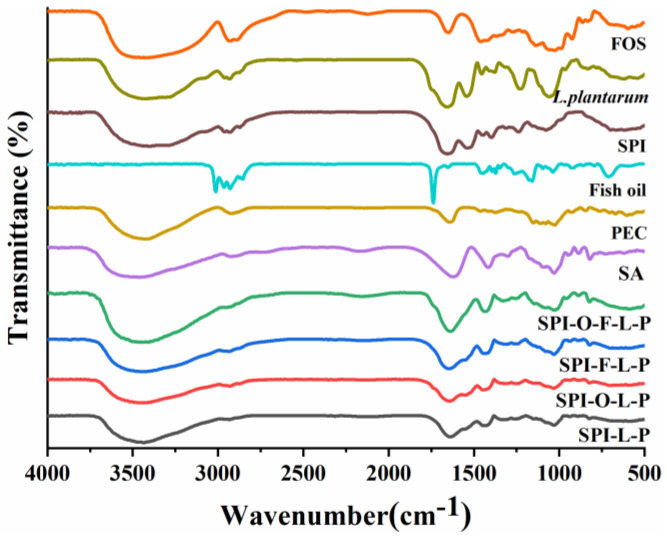
FTIR spectra of different samples.

**Figure 5 foods-11-03872-f005:**
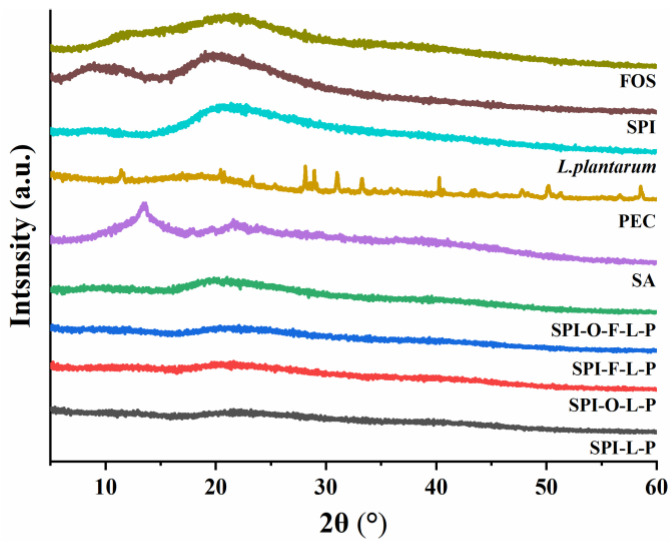
XRD pattern of different samples.

**Figure 6 foods-11-03872-f006:**
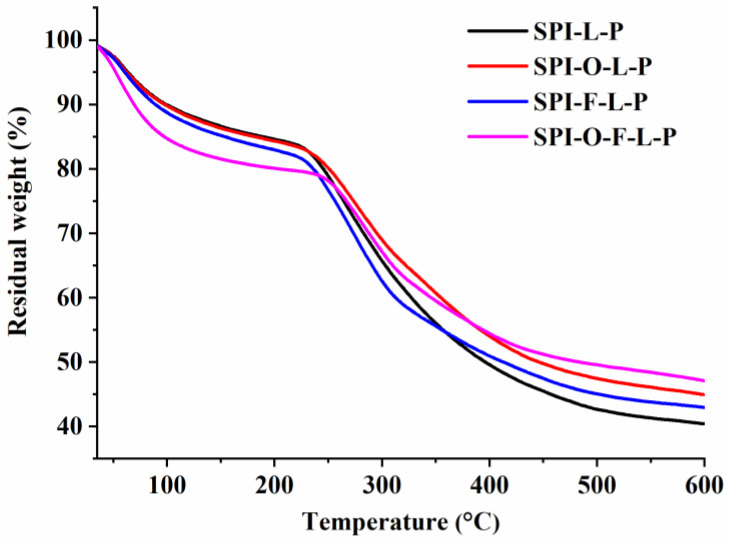
TGA curve of different samples.

**Figure 7 foods-11-03872-f007:**
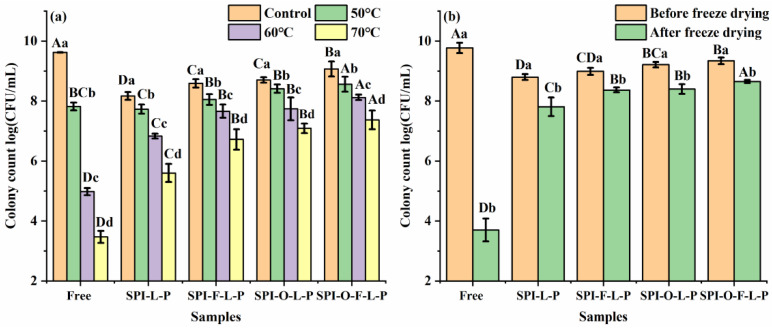
Resistance of different samples to heat treatment (**a**) and freeze drying (**b**). Different uppercase letters indicate significant differences between samples under the same treatment conditions and different lowercase letters represent significant differences between different treatment conditions for the same samples (*p* < 0.05).

**Table 1 foods-11-03872-t001:** The viability of probiotics encapsulated in different microcapsules during storage.

Storage at 4 °C					
Time (Days)	Free	SPI-L-P	SPI-F-L-P	SPI-O-L-P	SPI-O-F-L-P
0	9.36 ± 0.17 ^aA^	8.32 ± 0.23 ^aC^	8.70 ± 0.27 ^aBC^	8.77 ± 0.27 ^aB^	8.82 ± 0.22 ^aB^
15	7.15 ± 0.21 ^bB^	7.22 ± 0.28 ^bB^	8.15 ± 0.11 ^bA^	8.29 ± 0.12 ^bA^	8.36 ± 0.06 ^bA^
30	7.08 ± 0.08 ^bC^	7.13 ± 0.08 ^bcC^	7.74 ± 0.11 ^cB^	7.88 ± 0.19 ^cAB^	8.01 ± 0.14 ^cA^
60	5.89 ± 0.14 ^cD^	7.08 ± 0.06 ^bcC^	7.63 ± 0.23 ^cB^	7.82 ± 0.13 ^cAB^	7.98 ± 0.09 ^cA^
90	4.70 ± 0.24 ^dD^	6.90 ± 0.15 ^cC^	7.54 ± 0.13 ^cB^	7.77 ± 0.08 ^cAB^	7.92 ± 0.21 ^cA^
**Storage at 25 °C**					
**Time (Days)**	**Free**	**SPI-L-P**	**SPI-F-L-P**	**SPI-O-L-P**	**SPI-O-F-L-P**
0	9.36 ± 0.17 ^aA^	8.32 ± 0.23 ^aC^	8.70 ± 0.27 ^aBC^	8.77 ± 0.27 ^aB^	8.82 ± 0.22 ^aB^
15	6.89 ± 0.16 ^bB^	7.08 ± 0.31 ^bB^	7.44 ± 0.08 ^bA^	7.77 ± 0.19 ^bA^	7.78 ± 0.26 ^bA^
30	5.75 ± 0.10 ^cC^	6.17 ± 0.16 ^cBC^	6.38 ± 0.21 ^cA^	6.04 ± 0.18 ^cB^	6.27 ± 0.27 ^cA^
60	4.86 ± 0.10 ^dC^	5.75 ± 0.21 ^dB^	6.09 ± 0.07 ^dA^	5.81 ± 0.13 ^cB^	6.13 ± 0.09 ^cA^
90	2.49 ± 0.09 ^eC^	5.38 ± 0.11 ^eB^	5.73 ± 0.13 ^eA^	5.32 ± 0.15 ^dB^	5.68 ± 0.12 ^dA^

Note: Different lowercase letters on columns denote the significant difference between different storage days for the same samples (*p* < 0.05). Different uppercase letters on rows indicate the significant difference between samples at the same storage days (*p* < 0.05).

## Data Availability

Data are contained within the article.

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
