# Peer review of "Insights into Protective Effects of Different Synbiotic Microcapsules on the Survival of Lactiplantibacillus plantarum by Electrospraying"

_foods, 2022, doi:10.3390/foods11233872_

Round 1

Reviewer 1 Report

In this study Li and co-workers investigated the effect of different type of prebiotics on the survival of encapsulated prebiotics. The microcapsules separately containing fructooligosaccharide, fish oil and the complex of both evaluated based on the capsulation efficiency and survival rate after in vitro digestion, structure moreover the viability of the cells also tested under environmental stresses (heat treatment, freeze-drying and during storage). The study contains valuable and novel scientific data regarding the applicability of the encapsulated probiotic. The work is definitely fit to scope of journal and constructed correctly. The results presented in a well-structured manner. The data interpreted appropriately and consistently throughout the manuscript.

Some general and specific comments:

In my opinion, there are too much abbreviations in the article which are not always necessary, mainly in the abstract and conclusion chapter. It doesn't make it easier to understand.  In the abstract - Line 15 – This is the first time that the abbreviations SPI-F-L-P, SPI-O-L-P, and SPI-O-F-L-P appear and the meanings are unknown.

Line 99  Preparation of polymer solutions – this chapter is incomplete, we have no information on how  the capsule with fish oil and mixed prebiotic was prepared, what was the concentration (range). These parameters are in the Abstract, and some in the Results and Discussion but it would be better to collect/perform these information in Materials and methods section.

Line 226   Fig. 2 (c, d)  - actually it is Fig. 1.

Fig 2. – It is confusing, that ’A’ below the column and ’A’  above the column have different meanings. I suggest that using the abbreviation of the sample name on x axis instead of ABCD.   

Line 310 – ’Moreover, the absorption peaks of SPI-O-F-L-P were the sharpest among the microcapsules, indicating that the most probiotics was encapsulated in the SPI-O-L-P’.  Uncleared: SPI-O-L-P or SPI-O-F-L-P?

Reviewer 2 Report

I reviewed the manuscript entitled, Insights into protective effects of different prebiotics on the survival of encapsulated probiotics by electrospraying. The study has no novelty. Many studies in literature discussed on electrospraying of Lactobacillus plantarum. For example,

https://doi.org/10.1016/j.jff.2016.03.036

https://doi.org/10.3934/matersci.2016.1.114

The study objectives are well studies parameters in the literature

Methodology and study approach is very well known and shows no novelty

Reviewer 3 Report

The paper of Li et al. is a study of combining FOS and Fish Oil on survival of encapsulated probiotics produced by electrospraying. First of all, the authors must clarify whatever compounds have a Prebiotic affect (well known for FOS) or not (the statement is unclear for Fish Oil).

- Abstract and so on: For EE and SR (%), are 2 decimals significant? The reviewer recommends lo limit to 1 significant decimal.

- Introduction:

The authors classified Fish Oil as Prebiotic. The reviewer wonders if why Fish Oil could be classified as Prebiotic, as defined by International Scientific Association for Probiotics and Prebiotics (https://doi.org/10.1016/j.foodres.2021.110629). The authors refer to pub 15, but this one is focused on Polyphenols and on protective effects, and ref 16 (published by the same authors as those of the present paper). Fish oil may have protective effects without being a Prebiotic;

L63: Use the new name: Lactiplantibacillus plantarum;

- Materials and Methods:

L81: The strain should be available in a referenced collection;

L113: Please specify diameter for cal 17 and cal 21 syringes;

- Results and Discussion

L212: Use the new name: Lacticaseibacillus rhamnosus;

L291: Please, correct exponents;

L330: Does Legend of Fig5: “XRD pattern of different samples” refers to Diffraction angle? In this case, it must be included in legends of figure.

Reviewer 4 Report

GENERAL COMMENTS

The present study aims to evaluate the effects of different prebiotics on the survival of encapsulated Lactobacillus plantarum (recently named Lactiplantibacillus plantarum) obtained by electrospraying.  The manuscript is well designed and written and should be of interest to the readers to FOODS, also the chosen analytical techniques are good. However, I would recommend performing a language check by a native speaker.

In order to improve the clarity and the completeness of the work same modifications should be taken into account. Below are listed the major concerns.

TITLE:

In the title and throughout the manuscript I suggest changing "encapsulated probiotics" to "encapsulated Lactiplantibacillus plantarum".

There has been an extensive taxonomic restructuring in the of the family Lactobacillaceae so I suggest the authors to modify, throughout the manuscript, Lactobacillus plantarum with the recent name Lactiplantibacillus plantarum (see doi https://doi.org/10.1099/ijsem.0.004107).

MATERIALS AND METHODS:

Line 81: Detailed information should be provided on the strain of Lactiplantibacillus plantarum used in the study. Why only one Lactiplantibacillus plantarum strains? How was it selected? Based on what? How can the authors be sure to have taken into account the best strain to validate the system? In work like this more strains are used for two main reasons: i) to discriminate about species (different strains of different species); ii) to validate the results at species level (more strains of the species showing the best results). Of course, I am not asking to perform other tests, but please discuss about this point.

Line 82: Please include in the text the meaning of the acronym “MRS”.

Line 135 and Line 174: I suggest modifying the title of paragraphs by limiting the results of the study only to the strain of L. plantarum used.

Line 137: I suggest the authors add that incubation at 37°C for 24 h was conducted in anaerobiosis.

RESULT AND DISCUSSION

Line 198 and Line 345: I suggest modifying the title of paragraph by limiting the results of the study only to the strain of L. plantarum used.

Line 230: sample instead group

Line 241: difference instead change

Line 393: I suggest you add the appropriate reference: Succi et al. (Lactic Acid Bacteria in Pharmaceutical Formulations: Presence and Viability of “Healthy Microorganisms”. Journal of Pharmacy and Nutrition Sciences, 2014, 4, 66-75).

Round 2

Reviewer 2 Report

The study has no novelty. Many studies in literature discussed on electrospraying of Lactobacillus plantarum. Moreover, the study objectives are well studies parameters in the literature. Based on no novelty and weak scientific standards, I must recommend rejection.  Authors must consider addressing novel research hypothesis to contribute to the field of microencapsulation.

Author Response

.

Reviewer 3 Report

The manuscript has been significantly improved and is now suitable for publication.

Author Response

Thank you.